

# An energetics-based honeybee nectar-foraging model used to assess the potential for landscape-level pesticide exposure dilution

Johannes M. Baveco[1], Andreas Focks[1], Dick Belgers[1], Jozef J.M. van der Steen[2], Jos J.T.I. Boesten[1] and Ivo Roessink[1]

[1] Alterra, Wageningen University and Research, Wageningen, The Netherlands
[2] Plant Research International, Wageningen University and Research, Wageningen, The Netherlands

Corresponding author
Johannes M. Baveco,
hans.baveco@wur.nl

## ABSTRACT

Estimating the exposure of honeybees to pesticides on a landscape scale requires models of their spatial foraging behaviour. For this purpose, we developed a mechanistic, energetics-based model for a single day of nectar foraging in complex landscape mosaics. Net energetic efficiency determined resource patch choice. In one version of the model a single optimal patch was selected each hour. In another version, recruitment of foragers was simulated and several patches could be exploited simultaneously. Resource availability changed during the day due to depletion and/or intrinsic properties of the resource (anthesis). The model accounted for the impact of patch distance and size, resource depletion and replenishment, competition with other nectar foragers, and seasonal and diurnal patterns in availability of nectar-providing crops and wild flowers. From the model we derived simple rules for resource patch selection, e.g., for landscapes with mass-flowering crops only, net energetic efficiency would be proportional to the ratio of the energetic content of the nectar divided by distance to the hive. We also determined maximum distances at which resources like oilseed rape and clover were still energetically attractive. We used the model to assess the potential for pesticide exposure dilution in landscapes of different composition and complexity. Dilution means a lower concentration in nectar arriving at the hive compared to the concentration in nectar at a treated field and can result from foraging effort being diverted away from treated fields. Applying the model for all possible hive locations over a large area, distributions of dilution factors were obtained that were characterised by their 90-percentile value. For an area for which detailed spatial data on crops and off-field semi-natural habitats were available, we tested three landscape management scenarios that were expected to lead to exposure dilution: providing alternative resources than the target crop (oilseed rape) in the form of (i) other untreated crop fields, (ii) flower strips of different widths at field edges (off-crop in-field resources), and (iii) resources on off-field (semi-natural) habitats. For both model versions, significant dilution occurred only when alternative resource patches were equal or more attractive than oilseed rape, nearby and numerous and only in case of flower strips and off-field habitats. On an area-base, flower strips were more than one order of magnitude more effective than off-field habitats, the main reason being that flower strips had an optimal location. The two model versions differed in the predicted number of resource patches exploited over the day, but mainly in landscapes with numerous small resource patches. In landscapes consisting of few large

resource patches (crop fields) both versions predicted the use of a small number of patches.

## INTRODUCTION

There is serious concern about the widespread decline of pollinators in the agricultural landscape (*Potts et al., 2010*; *Vanbergen et al., 2013*). Agricultural intensification leading to increased stress from pesticides and lack of floral resources, is probably the main cause (*Goulson et al., 2015*). Honeybees, by exploiting mass-flowering crops, operating over long distances, and by being carefully managed by beekeepers, can be considered to be relatively insensitive to the disappearance and deterioration of semi-natural elements providing food and nesting opportunities (*Ricketts et al., 2008*; *Sponsler & Johnson, 2015*; *Winfree et al., 2009*). Their dependence on cropland for nectar and pollen acquisition, however, leads to high potential exposure to pesticides (*Krupke et al., 2012*).

Understanding to what extent pesticides may affect honeybees requires as an essential first step the quantification of their potential exposure. Exposure at the hive is the outcome of foraging in the landscape surrounding the hive, as concentrations of pesticides in nectar and pollen brought by foraging bees will depend on the provenance of these food items, and thus on choice of forage and foraging locations (*Garbuzov et al., 2015*). It is well known that honeybees may forage over long distances (*Beekman & Ratnieks, 2000*), up to approximately 10 km away from the hive. However, when sufficient high-quality resources are available nearby, most foraging will take place within a few kilometres (*Couvillon et al., 2015*; *Couvillon, Schurch & Ratnieks, 2014b*).

Models considering foraging in complex, heterogeneous landscapes with multiple resources to choose between, can be used to predict exposure concentrations at the bee hive. Such models may also serve as tools to test landscape management measures aimed at lowering exposure risk (mitigation). However, of the numerous honeybee models reviewed in *Becher et al. (2013)* and *Schmickl & Crailsheim (2007)*, none were developed to address landscape-level foraging and its consequences for pesticide exposure.

We extend and adapt the energetics-based modelling approach (*Cresswell, Osborne & Goulson, 2000*; *Dukas & Edelstein-Keshet, 1998*; *Schmid-Hempel, Kacelnik & Houston, 1985*) to deal with foraging for nectar on a large spatial scale, in heterogeneous landscapes with multiple resources including mass-flowering crops and wild flowers in semi-natural habitats. The model simulations can use real-world geographical information system (GIS) data at regional to national scales. Our model predicts resource patch selection during the day, taking into account resource depletion. It translates this selection into pesticide concentration in nectar arriving at the hive, when some patches are fields treated with pesticides or off-field habitats exposed to spray-drift. With the model, we explore under

which conditions the presence of multiple resources can divert foraging effort away from pesticide-treated fields and lead to dilution of exposure: a possible explanation of lower observed "field-realistic" rates of exposure compared to laboratory studies (*Carreck & Ratnieksi, 2014*).

We finally explore whether landscape management aimed at pollinator conservation may have the additional benefit of lowering the exposure of honeybees to pesticides. Three hypothetical scenarios were tested for their impact on exposure dilution, through the presence of attractive alternative crop fields, flowers strips (off-crop in-field resources), or attractive resources on off-field habitats.

## MATERIAL & METHODS

### Energetics-based foraging model

The model simulates honeybee nectar foraging over a single day in hourly time steps. The model ignores in-hive colony dynamics and assumes the colony to have limited to perfect knowledge of available resources and to adapt quickly to environmental fluctuations in food conditions (*Beekman & Lew, 2008*; *Beekman, Oldroyd & Myerscough, 2003*). Resource patch selection is based on net energetic efficiency. We differentiate between two basic model versions. In the "single-optimal" (SO) version of the model we assume that the colony's self-organization is fast enough to effectively focus on the single most-profitable food patch each hour. In case of two or more approximately equally profitable sources, mechanisms should be present that reinforce the use of only one of them (e.g., symmetry breaking and cross inhibition (*De Vries & Biesmeijer, 2002*)). In the "recruitment-limited" (RL) version we acknowledge that the rate of recruitment of foragers for a resource limits the speed with which the system can adjust to dynamic resources (*Seeley & Visscher, 1988*). As a consequence, in this version several resource patches with a high net energetic efficiency can be exploited simultaneously, as suggested by field observations (*Beekman et al., 2004*; *Visscher & Seeley, 1982*). In the SO-model this may occur when resource availability changes during the day, depending on the anthesis of flowers and/or due to resource depletion. In the RL-model, simultaneous exploitation of multiple patches during the day is the rule.

The model predicts for a hive at a specific location the set of resource patches used during a single day as well as several quantities that can be compared to field observations, e.g., the amount of sugar arriving at the hive, exploited nectar sources weighted by the amount of collected sugar in each, or the distribution of foraging distances. Alternatively, it allows for the quantification of impact of the hive population on a crop, e.g., as the number of flower visits per unit area, per flower or per patch.

### *Landscape and resources*

Foraging takes place in a landscape consisting of multiple resource patches. Each resource patch is assumed to be internally homogeneous and to contain a single resource. Resource patches may be fields with mass-flowering crops providing nectar or semi-natural habitats characterised by a dominant nectar-providing species.

Resources have a dynamic density of nectar-providing flowers $F$ (m$^{-2}$). In absence of nectar consumption, $F$ remains $F_0$. The density of open flowers is the product of plant

density and the number of open flowers per plant. Resources are characterised by the average amount of nectar $g$ (mg) a honeybee can obtain from a flower, and by the typical sugar content of its nectar (g g$^{-1}$).

### Resource patch selection

Being a central-place forager, a honeybee foraging trip comprises the travel back and forth between hive and resource patch, and the searching for and handling of flowers in the patch. Within the field, floral resources are collected assuming a type II functional response (*Holling, 1959*), so the number of flowers with nectar that are visited per time unit (s) amounts to

$$f = \frac{aF}{1+ahF} \qquad (1)$$

with $a$ the attack rate (m$^2$ s$^{-1}$) and $h$ the handling time per flower (s). The rate of nectar collection (mg s$^{-1}$) is given by $fg$. The time $t_L$ it takes to collect nectar to full capacity is

$$t_L = \frac{\gamma}{fg} \qquad (2)$$

where the capacity is given by $\gamma$ (mg). Note that unlike e.g., *Schmid-Hempel, Kacelnik & Houston (1985)* we assume that foragers always collect a full load.

Flight time and flight costs are proportional to the distance from hive to the resource patch. The duration of a foraging trip equals the sum of travel times and the time spent at the patch:

$$t_{\text{trip}} = 2\frac{D}{v} + \frac{\gamma}{fg} \qquad (3)$$

with $D$ being the distance from the hive to the field (m), and $v$ the flight velocity (m s$^{-1}$).

The energy expenditure $EE$ (J) ignoring basic metabolism can be calculated as the sum of the travel costs $EE_{\text{travel}}$ (travel time × energetic costs per time unit) and the costs while loading nectar in the field $EE_{\text{field}}$. The latter term is made up by flight costs while searching for nectar flowers and costs while sitting on the flowers extracting the nectar. The latter is ignored, as it is approximately an order of magnitude smaller than flight costs (a value of 0.0042 J s$^{-1}$ was applied in *Schmid-Hempel, Kacelnik & Houston (1985)*). Energy expenditure at the field thus becomes:

$$EE_{\text{field}} = \left(t_L - \frac{\gamma}{g}h\right)e_F = \frac{\gamma e_F}{gaF}. \qquad (4)$$

The equation is simplified by substituting Eqs. (2) and (1), thus eliminating the handling time. By assuming that handling the flower doesn't take energy, $h$ does not play a role in the energetic balance. In Eq. (4) average flight cost $e_F$ (J s$^{-1}$) is used, as during foraging in the field the individual state changes gradually from unloaded to loaded. The value of $e_F$ is obtained from loaded $e_{F,L}$ and unloaded $e_{F,U}$ flight costs (*Seeley, 1986*). The total travel costs are given by:

$$EE_{\text{travel}} = \frac{D}{v}\left(e_{F,U} + e_{F,L}\right) = 2\frac{D}{v}e_F. \qquad (5)$$

If the same route is followed back and forth, the energy costs can be averaged.

The total energy expenditure $EE_{\text{total}}$ (J) for a foraging bout sums to:

$$EE_{\text{total}} = \left(t_L - \frac{\gamma}{g}h\right)e_F + 2\frac{D}{v}e_F = \left(t_L - \frac{\gamma}{g}h + 2\frac{D}{v}\right)e_F = \left(\frac{\gamma}{gaF} + 2\frac{D}{v}\right)e_F. \qquad (6)$$

The yield of a trip in terms of energy, energy intake $EI$ (J), depends on the energy content of the collected nectar of resource type $R$, $e_R$ (J mg$^{-1}$):

$$EI = \gamma e_R. \qquad (7)$$

With $t_{\text{trip}}$, $EE$ and $EI$, the basic ingredients for a "decision-making process" for a colony are specified, and costs (both in energy and time) and yields for specific foraging locations in a landscape with multiple fields can be compared. In theory, there are different currencies that might be optimized (*Stephens & Krebs, 1987*). For honeybees, considerable effort has been invested in deciding whether the relevant currency is the net rate of energy delivery ((gain − cost)/time) or the net energetic efficiency ((gain − cost)/cost). Experimental data, e.g., *Seeley (1994)* but see *De Vries & Biesmeijer (2002)*, and models fit to experimental data (*Schmid-Hempel, Kacelnik & Houston, 1985*), as well as several of the other foraging models discussed by *Becher et al. (2013)* indicate that net efficiency is the most appropriate currency. We therefore assume that net energetic efficiency $NEE$

$$NEE = \frac{EI - EE_{\text{total}}}{EE_{\text{total}}} \qquad (8)$$

defines the attractiveness of each patch. In the SO version, the patch with maximum $NEE$ is selected as the single resource patch. In the RL version, the recruitment of foragers for a resource patch is explicitly modelled, following *Camazine & Sneyd (1991)* and *Seeley, Camazine & Sneyd (1991)* and extending their model in a somewhat similar way as done by *Cox & Myerscough (2003)*. In the RL-model, $NEE$ is translated into the probability of foragers abandoning resources and recruiting new foragers among the follower-bees present at the dance floor (Supplemental Information 2).

### Resource detection

Resource patches are detected primarily by scouts. Scouts make up around 10% of the unemployed foragers: novice foragers and experienced foragers that have recently abandoned a depleted resource patch (*Seeley, 1995*). Detection of an attractive resource patch may depend on patch distance and size (*Dauber et al., 2010*), as well as on the number of active scouts. Encounter rates may be obtained from movement models or from dispersal functions summarizing movement simulations (*Heinz et al., 2007*) as $\sigma_i$ values: the probability that a single scout encounters patch $i$. The probability for patch $i$ being detected by at least 1 of $s$ scouts amounts to

$$P_i = 1 - (1 - \sigma_i)^s. \qquad (9)$$

When $P$ is set to 1 for each resource patch–as we do in the studied cases–the hive population is assumed to have perfect knowledge of its environment. This setting is appropriate when

dealing with a landscape with abundant resources in the immediate surroundings of the hive (*Seeley, 1995*).

### Resource acquisition

The colony's hourly acquisition of a selected resource depends on the absolute number of foragers $n$ dedicated to this resource. For the SO-model, $n$ equals the total number of foragers active in a time step. For the RL-model, $n$ is a fraction of this total number, as multiple patches may be exploited simultaneously. Foragers may make several trips to the same patch, depending on trip duration $t_{\text{trip}}$ and the time between trips $t_{UD}$ spent on unloading and dancing in the hive. Thus, the number of foraging trips per hour an individual forager can make, amounts to $b = \Delta t/(t_{\text{trip}} + t_{UD})$ with $\Delta t$ representing the time step in seconds.

Not the full load of nectar $\gamma$ collected in the patch will arrive at the hive as some of the sugar will be consumed on the way back. The amount of energy used during the return flight, $E_c$, is approximately $e_L \frac{D}{v}$. The amount of nectar arriving at the hive thus decreases to $\gamma - \frac{E_c}{e_R}$ per trip and when summed over all foragers exploiting the patch during this time step becomes:

$$n\left(\gamma - \frac{E_c}{e_R}\right)b. \tag{10}$$

### Resource dynamics

By calculating resource patch selection per hour we may account for: (1) any diurnal pattern in the anthesis of flowers; (2) depletion of floral resources occurring in small resource patches; and (3) diurnal patterns in the number of active foragers in the hive. For simplicity's sake, we assume a binary pattern in anthesis (all flowers either open or closed).

Resource dynamics may also result from nectar consumption by competing species or foragers from other colonies. A constant background density of competitors, $Z$, may be defined of species that exploit resources in a similar way as honeybees (same functional response). The main reason to incorporate competition is that it allows us to explore honeybee foraging in a setting of quickly depleting resources.

Within a day there may be also renewal of nectar resources. With a non-zero renewal rate, $r$, a fraction of the pool of visited (empty) flowers will each time step turn into nectar-providing flowers again.

We deal with resource depletion by subtracting after each time step the visited number of open flowers from the initially present number. Hourly dynamics in open flower density $F$ are given by:

$$F_{t+1} = F_t - \frac{n\gamma}{gA}b - \frac{aF_t}{1+ahF_t}Z + r(F_0 - F_t) \tag{11}$$

with $A$ representing field size (m$^2$). A fixed number of flowers $\gamma/g$ is visited for a full load. Lowered flower density may change the *NEE*-based ranking of resource patches in the next time step.

## Exposure Assessment

When the foraging model is combined with information on fields being treated with a chemical, and/or off-field habitats being exposed to spray-drift, the concentration of the chemical in the nectar brought into the hive can be estimated. It depends on the concentration of the chemical in the nectar of flowers in exploited patch $i$, $C_i$, expressed in e.g., µg mg$^{-1}$. When the chemical is not metabolized together with the sugar on the way back to the hive, its total amount (mg) arriving at the hive will be:

$$\sum_{i=1}^{L} n_i \gamma \, b_i C_i. \tag{12}$$

The summation is made over $L$, the set of all exploited patches over all hours. The concentration of the chemical in nectar at arrival no longer equals $C_i$ but becomes $C_i / \left(1 - \frac{E_{c,i}}{\gamma e_{R,i}}\right)$, implying enrichment. When the chemical is metabolized together with the sugar, its concentration in the nectar remains the same but the amount arriving at the hive will be lower:

$$\sum_{i=1}^{L} n_i \left(\gamma - \frac{E_{c,i}}{e_{R,i}}\right) b_i C_i. \tag{13}$$

From the point of view of exposure risk inside the hive, the relevant chemical concentration should be expressed on a sugar base: nectar with a low sugar content will be concentrated until a minimum sugar content is reached. The exposure on a sugar-base is obtained by dividing the total amount of the chemical (µg) entering the hive by the total amount of sugar entering the hive (mg). Without metabolization of the chemical (with enrichment) we obtain as sugar-based daily-averaged concentration of the chemical (µg mg$^{-1}$) entering the hive:

$$X_e = \frac{\sum_{i=1}^{L} n_i \gamma \, b_i C_i}{\sum_{i=1}^{L} n_i \frac{(\gamma e_{R,i} - E_{c,i})}{e_{\text{SUGAR}}} b_i}. \tag{14}$$

When the chemical is metabolized with the sugar (no enrichment) we obtain:

$$X_m = \frac{\sum_{i=1}^{L} n_i (\gamma - \frac{E_{c,i}}{e_{R,i}}) b_i C_i}{\sum_{i=1}^{L} n_i \frac{(\gamma e_{R,i} - E_{c,i})}{e_{\text{SUGAR}}} b_i}. \tag{15}$$

Equations (14) and (15) quantify the contribution of each selected resource patch to the sugar-based concentration of a chemical (µg chemical mg$^{-1}$ sugar) entering the hive. Using this information together with the calculated amount of sugar having this concentration we can construct detailed distributions quantifying the relative composition of the nectar entering a particular hive, in terms of sugar-based concentrations.

### Dilution

Exposure dilution is caused by foraging on other resources than a treated resource when these other resources are not or to a lesser extent contaminated. Dilution is defined relative

to a reference concentration in treated resources, e.g., the sugar-based concentration $X_P$ in the nectar of a treated field with resource $R$:

$$X_P = C_P \frac{e_{\text{SUGAR}}}{e_R} \tag{16}$$

where $C_P$ refers to the concentration (on a wet-weight base) in nectar on the treated field, often referred to as the predicted environmental concentration ($PEC$) resulting from a certain application rate of the chemical. It is multiplied by $e_{\text{SUGAR}}/e_R$ to obtain the sugar-based concentration. Dilution factors follow from actual $X_m$ or $X_e$ as:

$$\varphi_m = X_m/X_P \text{ and } \varphi_e = X_e/X_P. \tag{17}$$

With a patch-specific $C$ we can represent variability in the applied dose for the treated crop, deal with a substance that is applied on different crops with different doses, and include patches representing off-crop or off-field habitats that are exposed through spray drift only.

For an agricultural landscape containing fields with a target crop as the only resources, Eqs. (14) and (15) can be further simplified. With a single resource $e_{R,i}$ equals $e_R$ and when application of a pesticide on the target crop leads to a constant concentration $C_P$ in the nectar, $C_i$ equals $C_P$. Dilution obtained applying Eq. (14) then simplifies to:

$$\varphi_e = \frac{\sum_{i=1}^{L} n_i \gamma b_i C_P}{\sum_{i=1}^{L} n_i \frac{(\gamma e_R - E_{c,i})}{e_{\text{SUGAR}}} b_i} \frac{1}{C_P \frac{e_{\text{SUGAR}}}{e_R}} = \frac{\sum_{i=1}^{L} \delta_i n_i \gamma b_i}{\sum_{i=1}^{L} n_i (\gamma e_R - E_{c,i}) b_i} \tag{18}$$

with $\delta_i = 1$ in case the field selected in time step $i$ is sprayed and $\delta_i = 0$ if it is unsprayed. Similarly, for the case with metabolization and no enrichment (Eq. (15)) we obtain:

$$\varphi_m = \frac{\sum_{i=1}^{L} \delta_i n_i (\gamma - \frac{E_{c,i}}{e_R}) b_i C_P}{\sum_{i=1}^{L} n_i \frac{(\gamma e_R - E_{c,i})}{e_{\text{SUGAR}}} b_i} \frac{1}{C_P \frac{e_{\text{SUGAR}}}{e_R}} = \frac{\sum_{i=1}^{L} \delta_i n_i (\gamma e_R - E_{c,i}) b_i}{\sum_{i=1}^{L} n_i (\gamma e_R - E_{c,i}) b_i}. \tag{19}$$

Eqs. (18) and (19) show that for this simplified case, dilution factors do not depend on $C_P$ but simply reflect how many unsprayed fields are selected. When all selected fields are treated $\varphi_m$ will be one, while $\varphi_e$ may even exceed one. Whether selected resource patches were treated or not depends on the application scenarios. In a probabilistic approach, with $p$ representing the probability for a field of being treated, dilution will on average approximate $p$.

### 90th percentiles

In an exposure assessment the model will be applied for all possible locations of an apiary within the area of use of a substance. Following recent guidance (EFSA, 2013), all sites at edges of target crop fields are considered to be potential locations and the target crop patch adjacent to the hive location is always assumed to be sprayed or treated. From application of the model at each site, we construct cumulative distributions of exposure or dilution from which statistical descriptors can be obtained, e.g., the 90th percentile that quantifies the dilution obtained at 90% of all sites, or the exposure that is exceeded at 10% of the sites.

## Model Analysis & Application
### Energetics-based model
From the energetics-based model (Eqs. (6) and (7)) we derived thresholds for the exploitation of resources depending on distance and resource characteristics, as done by *Cresswell, Osborne & Goulson (2000)*. For resources at equal distance we derived a simple equation to compare their attractiveness. For landscapes consisting of only large-scale crop fields, we derived rules of thumb for the selection of foraging locations.

### Landscape representation
When it is assumed that nectar is collected from all resources in the landscape proportional to their attractiveness (*EFSA, 2013*), exposure dilution is always to be expected in landscape mosaics with multiple resource patches. In the RL-model dilution is more likely than in the SO-model. To explore whether landscape-level exposure mitigation can be effective, we tested for both models three scenarios, all based on the idea that by providing alternative floral patches foraging effort can be diverted away from exposed patches. Construction of flower strips and pollinator-friendly management of semi-natural habitats are such measurements that may contribute to the persistence of pollinator populations in the agricultural landscape (*Garibaldi et al., 2014*; *Haaland, Naisbit & Bersier, 2011*; *Wratten et al., 2012*). Also, the presence of different (early and late) mass-flowering crops has been suggested to enhance pollinator density (*Riedinger et al., 2014*).

We parameterized the model for two flower species (Table 2), oilseed rape (*Brassica napus*, OSR) as a common attractive mass-flowering crop, and white clover (*Trifolium repens*) being an important floral resource, common along roadsides and field margins (*Sponsler & Johnson, 2015*). Patches with these resources provided the building blocks of the landscapes tested in the scenarios. In all scenarios OSR represented the target crop. From the model it followed that for any of the scenarios to be effective the alternative resource had to be equally or more attractive (larger *NEE* at equal distance) than the OSR field. For the scenario study, using a hypothetical landscape, we selected for OSR a conservative and for clover an optimistic estimate of the relevant coefficients (Table 2 and Supplemental Information) and checked whether this condition was met (see 'Results'). In real landscapes clearly more resource types will be present, in a range of densities. To understand the impact of some of this variability we varied flower densities and sugar content over a range of values.

We defined landscape structure from geographic data for the northern part of Flevoland (The Netherlands). Geo-referenced data for crops were obtained from the land-use database LGN6 (grid-based, 25 m resolution) (*Hazeu et al., 2011*). Data on the presence of off-field habitats (road side verges, ditch sides, shores, other semi-natural elements) were obtained from the vector-based dataset TOP10NL (*PDOK, 2015*). We assumed that fields of LGN6 category "other crops" represented OSR. For the "Alternative Fields" scenario, fields of another (randomly chosen) LGN6 category represented the alternative mass-flowering crop (Fig. 1). For the "Off-field Habitats" scenario, off-field habitats were used as specified in the spatial data set. All sites (cells in the 25 m grid) that bordered target crop fields were considered as potential beehive locations (Fig. S1). To avoid excessive computing times
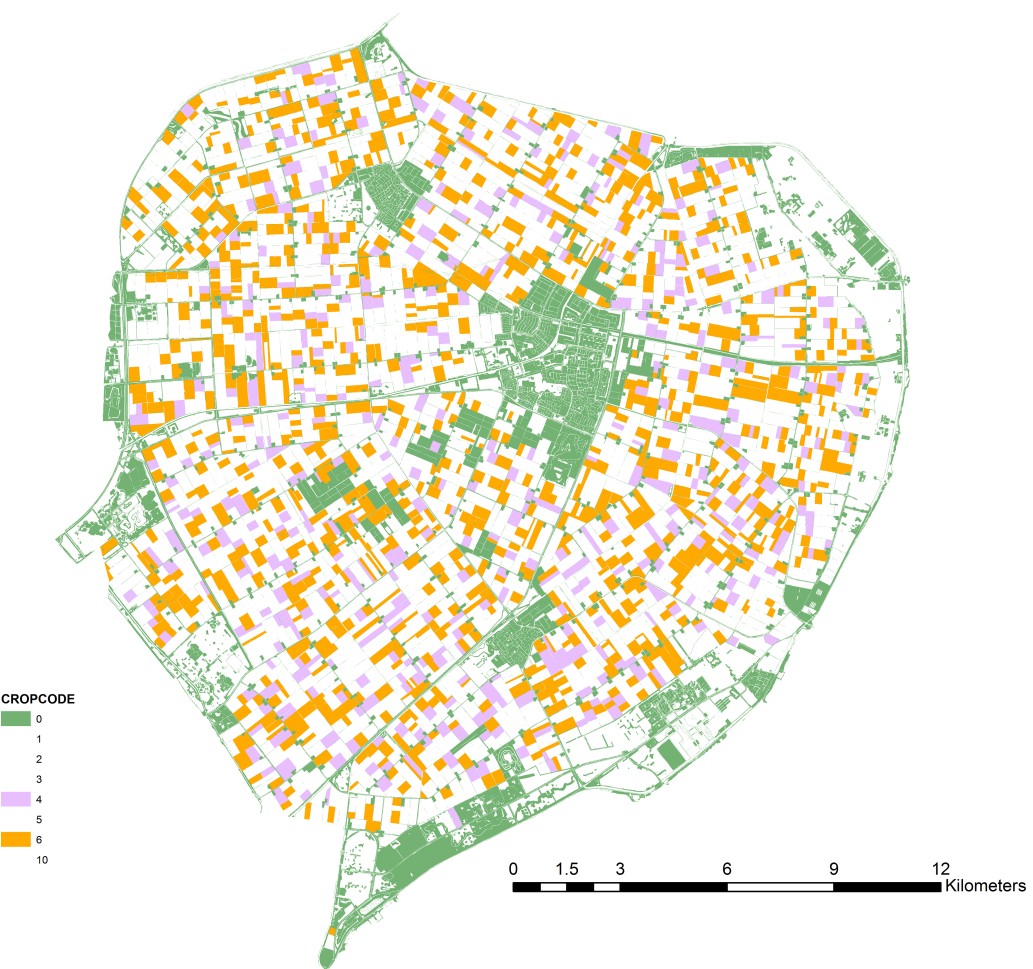

CROPCODE
- 0
- 1
- 2
- 3
- 4
- 5
- 6
- 10

**Figure 1  Map of the case-study area.** Oilseed rape fields (ochre), alternative crop fields (lilac) and off-field resource patches (green) as used in the scenario simulations.

the model was finally run on a random selection of 10% of these sites (identical for all scenarios and runs). Tested landscapes contained 1,207 fields with OSR (7,057 ha). The alternative crop was present on 454 fields (3,375 ha). Off-field habitat patches were small and numerous: 58,137 patches comprising 3,410 ha.

We assumed all resources to have open flowers during the whole day, a foraging day of 10 h, no competitors being present ($Z = 0$), no renewal of resources ($r = 0$), 100 active foragers and coefficients as in Tables 1 and 2. By definition, sites were next to a large resource patch. Therefore, considering only resources within 2 km distance ($D_{max} = 2000$) and assuming perfect knowledge of their presence ($P = 1$ for each resource patch) seemed reasonable. For simplicity's sake we assumed that the hypothetical pesticide was metabolized with the consumed sugar on the way back to the hive, so without dilution the dilution factor would simply be one. The nearby crop field was assumed to be always treated with the pesticide.

**Table 1  Fixed (energetic) coefficients for honeybee and nectar.**

| Coefficient | Symbol | Dimension | Value | Source |
|---|---|---|---|---|
| Maximum foraging distance | $D_{max}$ | m | 2000[a] | |
| Flying speed | $v$ | m s$^{-1}$ | 4.17[b] | *de Vries & Biesmeijer (1998)* |
| Flight cost (unloaded) | $e_U$ | J s$^{-1}$ | 0.037[c] | *Seeley (1986)* |
| Flight cost (loaded) | $e_L$ | J s$^{-1}$ | 0.075 | *Seeley (1986)* |
| Capacity (maximum load) | $\gamma$ | mg | 32.5 | *Winston (1987)* |
| Energetic value sugar | $e_{SUGAR}$ | J mg$^{-1}$ | 17.2 | *Seeley (1985)* |
| Time unloading nectar and dancing | $t_{UD}$ | s | 300[d] | *Seeley, Camazine & Sneyd (1991)* |

Notes.

[a] Specific for studied scenarios. Considerably longer distances have been reported, e.g. max. 13.5 km, 95% within 6 km, mean 2.3 km (*Beekman & Ratnieks, 2000*).

[b] Other reported value 6.95 m s$^{-1}$ (*Seeley, 1995*).

[c] Other reported value 0.0334 J s$^{-1}$ (*Schmid-Hempel, Kacelnik & Houston, 1985* and references therein).

[d] *Goodwin et al. (2011)* observed longer times in the hive between foraging (11.13 min). Range 150–300 in the model of *Seeley, Camazine & Sneyd (1991)*.

**Table 2  Resource-specific coefficients.** Coefficient values for the two resources considered in the study, oilseed rape and clover.

| Crop specific parameters | | | Oil-seed rape | Off-field (clover) |
|---|---|---|---|---|
| $g$ | Nectar per flower | mg | 1.071[a] | 0.373[b] |
| $F_0$ | Open flowers density | m$^{-2}$ | 264[c] | 2808[d] |
| | Share of sugar in nectar | g g$^{-1}$ | 0.47[e] | 0.74[f] |
| $e_R$ | Energetic content nectar | J mg$^{-1}$ | 8.084[g] | 12.728[h] |
| $a$ | Attack rate | m$^2$ s$^{-1}$ | 0.00147[i] | 0.00020[j] |
| $h$ | Handling time per flower | s | 4.1[k] | 1.22[l] |

Notes.

[a] Range 0.7–6 µl (average 2.0 µl) (*Pierre et al., 1999*), approximately $0.7 + 0.7 * 0.53 = 1.071$ to $6 + 6 * 0.53 = 9.18$ mg.

[b] Range 0.143–0.351 (10 °C) and 0.112–0.373 (18 °C) (*Jakobsen & Kritjánsson, 1994*).

[c] Own measurements: (June 2014: 1182; September 2015: 264).

[d] *Goodwin et al. (2011)* for clover field, defined as density of open florets ( =15.6 open florets per flower x 180 flowers m$^{-2}$). *Burdon (1983)*: 1500–3000 open flowers, derived from 20–40 florets per flower-head (average 30), assuming 50 –100 flower-heads per m$^2$.

[e] Range 0.47–0.59 compiled from *Pritsch (2007)* and *Maurizio & Grafl (1969)*.

[f] *Jakobsen & Kritjánsson (1994)*.

[g,h] From (share of sugar) * $e_{SUGAR}$.

[i,j] See Supplemental Information 1.

[k] *Free & Nuttall (1968)*.

[l] *Peat, Tucker & Goulson (2005)* for small bumblebee.

## Exposure mitigation

(i) **Alternative Fields:**  dilution can result from the presence of attractive but untreated fields, either with the target crop or with alternative attractive (mass-flowering) crops. For crop fields other than the one nearest to the hive, treatment occurred following a probability *p*. The average dilution factor for a site was then calculated from 100 different random

realizations of the series of treated fields. The analysis was done for a range of values for $p$ (0–1, with an interval of 0.1). A variant of this mechanism would be the presence of fields with another, attractive, crop type. We tested this by defining another common crop in Flevoland as a hypothetical alternative mass-flowering crop, largely identical to oilseed rape (Fig. 1). The energetic attractiveness of this crop was manipulated to range from less to more attractive than OSR, by adjusting its sugar content (0.8, 0.9, 1, 1.1 and 1.2 times OSR sugar content). Thus, the alternative crop could also represent another OSR variety.

(ii) **Flower Strips**: dilution can result from the presence of highly attractive flower strips around target crop fields. This was simulated by adding to each target crop field up to 4 flower strips at its edges, each strip of length 1/4 of field perimeter. Strips contained white clover (*Trifolium repens*, Table 2) in densities comparable to clover fields. Presence of each strip was randomly set, with a probability $p$ (0–1, with an interval of 0.1). The area of a flower strip was set to a prescribed width $w$ multiplied by strip length. Width values of 1, 2, 5 and 10 m were tested. The area contained in the strips was subtracted from the crop field area. Thus, strips were strictly in-field off-crop habitats.

(iii) **Off-field Habitats**: dilution can result from the presence of off-field semi-natural habitats when these are managed in an appropriate way. We tested this using geo-data for fields (as above) with OSR and for semi-natural elements (Fig. 1). We assumed a single common flower species (white clover) to be representative for all off-field habitats. As the presence and size of these habitats were fixed and defined by the geo-data, we tested their impact on dilution for a range of resource quality values. Open flower density determined quality and was set to low, medium or high value (0.5, 1 or 1.5 times clover field density). The three quality levels were tested for a range of values for $p$, here representing the probability that an off-field element was considered a nectar resource patch.

## RESULTS

### Energetics-based model

From the energetics-based model (Eqs. (6) and (7)) we derived thresholds for the exploitation of resources depending on distance to the resource patch and other resource characteristics. The energy balance ($EI - EE$) for a foraging trip has to be positive (*Dukas & Edelstein-Keshet, 1998*), leading to the condition:

$$\gamma e_R - \left( \frac{\gamma}{gaF} + 2\frac{D}{v} \right) e_F > 0. \tag{20}$$

The threshold distance at which a resource patch cannot be exploited any more (Fig. S2) amounts to

$$D_T = \frac{v\gamma}{2} \left[ \frac{e_R}{e_F} - \frac{1}{g}\left( \frac{1}{f} - h \right) \right] = \frac{v\gamma}{2} \left[ \frac{e_R}{e_F} - \frac{1}{gaF} \right]. \tag{21}$$

Resource patch selection based on net energetic efficiency implies selecting the patch for which the ratio between energy intake and expenditure *EI/EE* is largest:

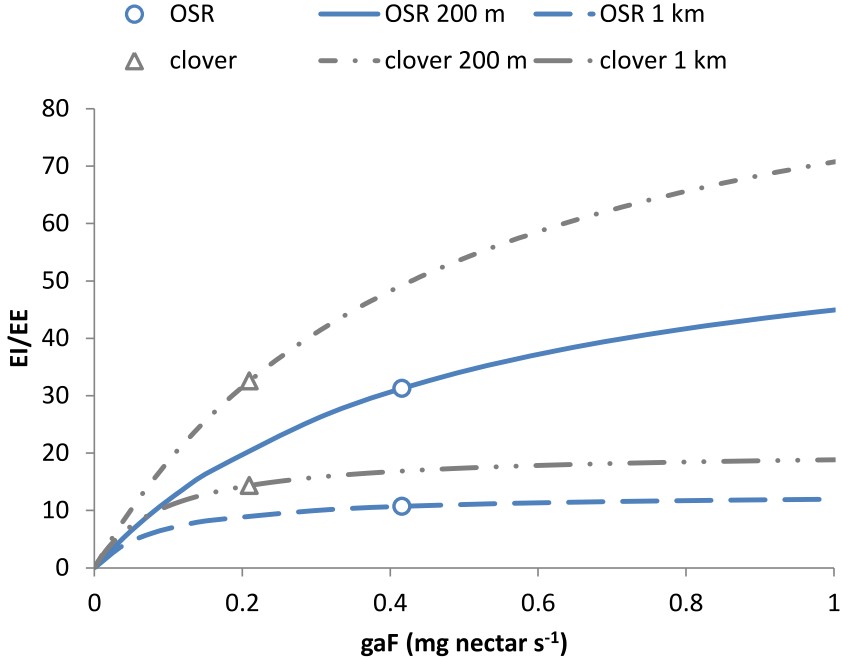

**Figure 2  Choice of resources depends on net energetic efficiency.** The choice between resources at equal distance depends on their value for *EI/EE* (Eq. (22b)). This ratio increases asymptotically with nectar acquisition rate *gaF* to the limit value $1/c \times e_R/e_F$. With larger $e_R$, e.g., for clover compared to OSR, *EI/EE* will level off at a higher value. When flower density is much higher, the resulting value of *EI/EE* may still be larger for the resource with the lower $e_R$. Curves refer to resources at 200 m and 1 km. Values for clover and OSR as specified in Table 2 and used in the scenarios are indicated.

$$\frac{EI}{EE} = \frac{\gamma\, e_R}{\left(\frac{\gamma}{gaF} + 2\frac{D}{v}\right)e_F} \tag{22}$$

This can be rewritten to:

$$\frac{EI}{EE} = \frac{gaF}{\left(1 + \frac{2D}{\gamma v}gaF\right)} e_R/e_F. \tag{22a}$$

Or, substituting $\frac{2D}{\gamma v}$ by constant *c* for patches at equal distance:

$$\frac{EI}{EE} = \frac{gaF}{\left(1 + cgaF\right)} e_R/e_F. \tag{22b}$$

Figure 2 shows how this ratio depends on resource characteristics *gaF,* the nectar acquisition rate. At equal distance, *EI/EE* scales linearly with energetic content $e_R$ and asymptotically with *gaF*. In a mass-flowering case, a further increase of *gaF* will not increase *EI/EE* much. For a "sparse-flowering" resource to compete in attractiveness with a mass-flowering crop, its energy content or flower density needs to be considerably higher.

An 'extreme' case results from landscapes consisting entirely of fields with mass-flowering crops. There, open flower density *F* will be very large, and the functional response *f* will approach $1/h$. As a consequence, the energy spent in the field $EE_{\text{field}}$ will be approaching

zero and *EE* will thus be determined mostly by $EE_{travel}$. The threshold distance (Eq. (21)) will simplify to a linear relationship with $e_R$, with steepness independent of other crop properties; the honeybee constants are given between brackets:

$$D_T \approx \left[ \frac{\gamma v}{2 e_F} \right] e_R. \tag{23}$$

Maximising the net energetic efficiency in the mass-flowering case means maximising

$$\frac{EI}{EE} \approx \left[ \frac{\gamma v}{2 e_F} \right] \frac{e_R}{D}. \tag{24}$$

For fields at equal distance, the selected field will thus be the one with the highest energy content $e_R$.

### Landscape scenarios

Cumulative distributions of dilution factors were discontinuous for the SO-model, with many sites experiencing no dilution at all, others having no exposure and some experiencing limited dilution (Figs. S3–S6). For the RL-model the distributions were continuous, with few sites being without exposure or without dilution. All distributions were summarized by their 10-, 50- and 90-percentiles (Fig. S6). For all scenarios the number of resource patches exploited over the day were obtained as well, as a characteristic output of the foraging model. For the RL-model, this referred to the number of resource patches accounting for 90% of the sugar brought to the hive.

### "Alternative Fields"

In this scenario the number of resource patches and resource area were constant. In two sub-scenarios either only OSR fields or OSR fields plus fields with a similar crop of different quality (sugar content) were present.

When fields with OSR were the only resource patches, a smaller (SO-model) or larger (RL-model) fraction of the sites showed considerable dilution when a fraction of these fields were not sprayed (Fig. S3). Even when none of the fields besides the nearest were treated there was no site completely without exposure, indicating that the nearest (by definition treated) field was always included in the set of resource patches exploited during the day. For the RL-model, there was always some dilution, especially when the fields selected besides the nearest field had a high probability of *not* being sprayed.

In the presence of an alternative untreated crop there was a fraction of sites for which an alternative crop field was included in the set of exploited patches, resulting in dilution (Fig. S4). For the SO-model, these sites had no exposure at all, implying that this alternative crop field constituted the single patch exploited during the day. Depletion played no role on large fields and either the nearest OSR field or an alternative crop field was selected and remained optimal during the whole day. For the RL-model, there was always some dilution, but there were no sites completely without exposure as the nearby OSR field was always part of the exploited patches set.

The number of exploited patches over the day was on average small: one for the SO-model and approximately 3–4 for the RL-model (Fig. S7). Thus the resource patch

chosen at the beginning of the day usually stayed the optimal one (SO-model) during the day, as depletion was unlikely on (large) fields. In this structurally simple landscape there were always only a few nearby attractive fields that were exploited simultaneously in the RL-model. With increased sugar content the distance over which the alternative crop was energetically attractive increased and thus the number of exploited patches became larger (RL-model).

### "Flower Strips"

In this scenario the number of resource patches increased with $p$, the probability of a flower strip being present at a side of a OSR field. Total resource area remained constant, but the ratio between the two resource types changed with $w$, strip width, as strips were located in-field.

For both models the fraction of sites with considerable dilution increased profoundly with $p$ (Fig. 3A and Fig. S5). For the SO-model (Figs. S5 and S6) increasing the width of strips increased the dilution reached by 90% of the sites. Wider strips could keep up a higher *NEE* than the OSR field over a longer time, as depletion in the wider strips was less likely. For the RL-model, impact of width was negligible: depletion was unlikely as foraging effort was already divided over multiple patches representing a relatively large area (including the OSR field).

For the RL-model the number of exploited patches (Fig. 4B) increased linearly with $p$: more strips being present in the neighbourhood implied more patches being used. For the SO-model (Fig. 4A) only with narrow strips the number of exploited patches per day increased with $p$. Narrow strips were quickly depleted and when more strips were present in the neighbourhood another strip could become the optimal patch of the next hour. Wider strips implied less depletion and strips being selected as optimal at the beginning of the day had a higher chance of remaining the single optimal patch during the day.

### "Off-field Habitats"

In this scenario the number of resource patches and resource area increased with $p$, the probability of an off-field habitat patch being managed for nectar resources of different quality (clover open flower density).

Presence of high quality off-field resource patches affected the distributions of dilution factors in a similar way as narrow flower strips (Figs. 3B–3D, Fig. S6). The dilution factor cumulative distribution was very sensitive to off-field resource quality. With medium quality less dilution was reached at all sites and for low quality hardly any off-field resource patches were selected at all.

The number of exploited patches showed a similar pattern as in the flower strips scenario (Figs. 4C and 4D). For the SO-model with high and medium quality off-field habitats, the exploited patch number increased with $p$ to 4 while for low quality it remained constant (1). For the RL-model, exploited patch number increased linearly with $p$ reaching a very high level (average of 75) with high quality off-field habitats.

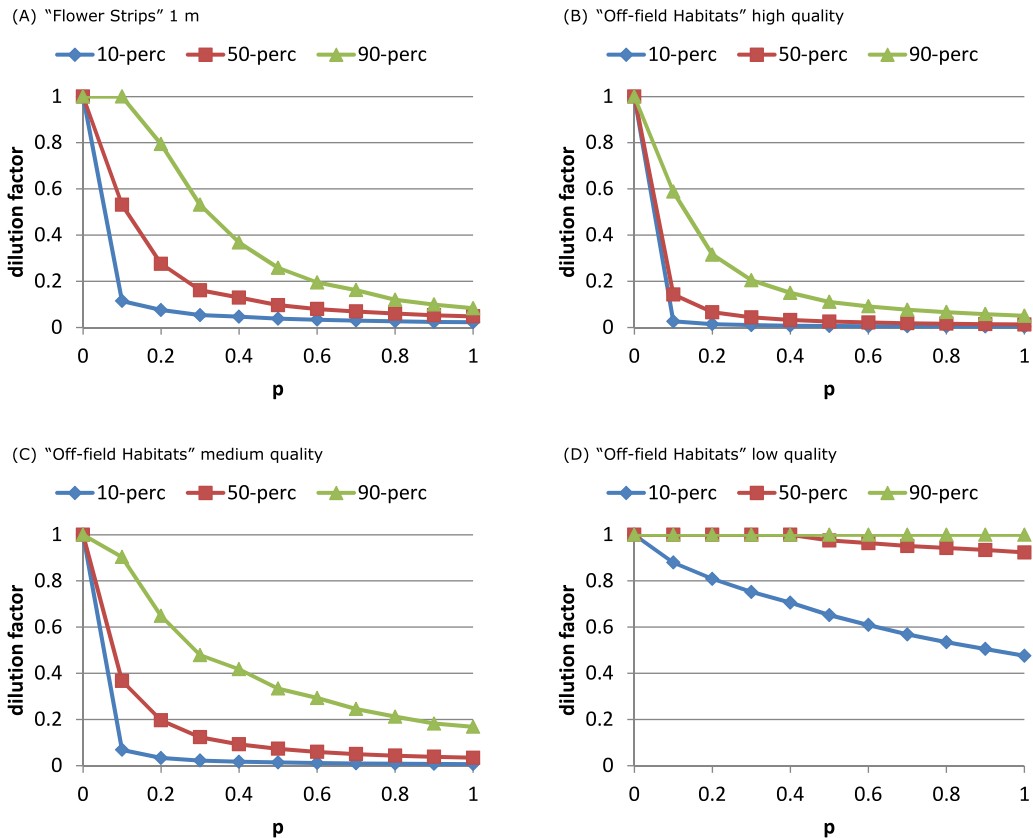

**Figure 3  Attractive flower strips and off-field habitats may dilute exposure.** Flower strips can lead to high exposure reduction, when present in large numbers (A). For the RL-model, the width of the strips has no impact on this dilution; for 2, 5 and 10 m wide strips, the graphs are very similar (not shown). Off-field habitats can have similar impact (B, C) as flower strips. When quality is low (D) few off-field habitat patches are selected resulting in little dilution. Note that flowers strips are identical to medium quality off-field habitat patches. All graphs refer to RL-model.

## Comparing scenarios

The "Flower Strips" scenario and "Off-field Habitats" medium quality scenario were similar as both created the same type of new resource patches, in-field versus off-field. We compared their effectivity by plotting the dilution percentiles against the total area covered by the patches. The "Flower Strips" scenario with narrow strips (1 m) appeared roughly 35 times as efficient as the "Off-field habitat" scenario (Fig. 5) as flower strips were in-field resource patches located always near the considered bee hive. Off-field resource patches on the other hand were not guaranteed to be sufficiently close to a target crop field and the bee hive site to have an impact.

## DISCUSSION & CONCLUSIONS

### Conceptual model

In the model the selection of a resource patch was based on energetic efficiency. This implies that the duration of all phases of a foraging trip, including handling time of the flowers, did not affect patch choice. Duration did however determine the number of trips that could

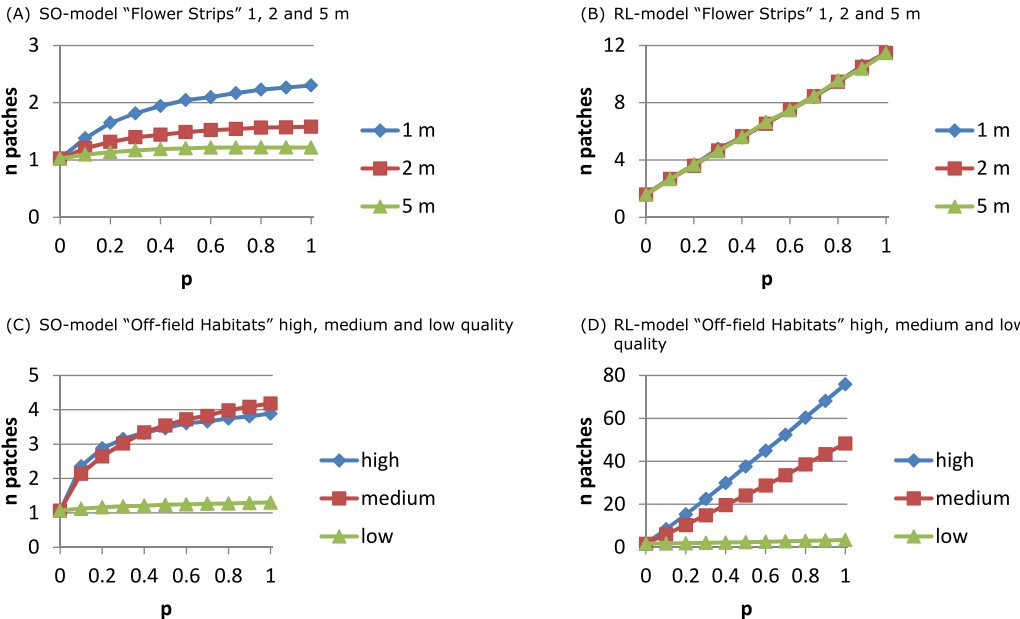

**Figure 4 The number of daily exploited patches depends on the foraging patch selection model.** With the SO-model an increase in the number of exploited patches indicates that depletion of the optimal patch plays a role. This may occur for narrow flower strips (A) and for off-field habitats that are mostly small-sized (C), and is determined by patch size not by its 'quality'. For the RL-model the number of exploited patches is linearly related to their abundance. Depletion is unlikely and patch size has no impact on patch selection (B). Instead patch 'quality' is important (D) as patches need to have a sufficiently high *NEE* to be considered and promoted for in the recruitment process.

be made, and thus the colony's rate of resource acquisition and of resource depletion. In a similar way, the size of a resource patch had no direct effect on its attractiveness, only indirectly through faster depletion of smaller patches. A smaller patch may also have a larger probability of remaining undetected (*Dauber et al., 2010*). With a fixed number of foragers exploiting a single patch (SO-model) or being distributed over a small set of patches (RL-model) the model predicted a decreasing flower visiting rate with patch size. This seems in accordance with findings of *Goulson (2000)*.

The maximum distance at which nectar-providing resource patches can be exploited is given by Eq. (21). Based on the values of Tables 1 and 2 the distances estimated for OSR and clover fields, 8.1 and 12.7 km, respectively, are well within the range of maximum observed foraging distances. For these mass-flowering crops, the threshold distance is mainly defined by the energy content of their nectar (Eq. (23)). For natural elements with sparser flower distributions than found on clover fields, maximum distances are likely lower (Fig. 2), as searching times will be higher, increasing the energetic costs of traveling between flowers.

## Resource patch selection

On a landscape level, the choice between resource patches depended on the relative value of their energetic efficiency. The attractiveness of different resources at different distances can be compared with each other using Eq. (22), hence exposure mitigation strategies based on the provision of alternative uncontaminated resource patches can be evaluated based

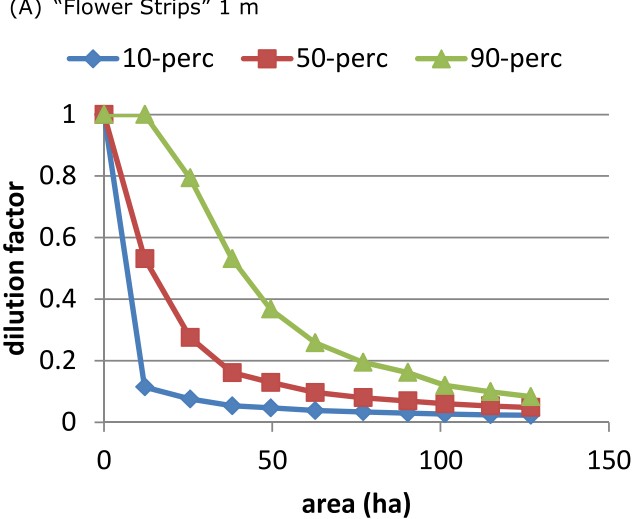

(A) "Flower Strips" 1 m

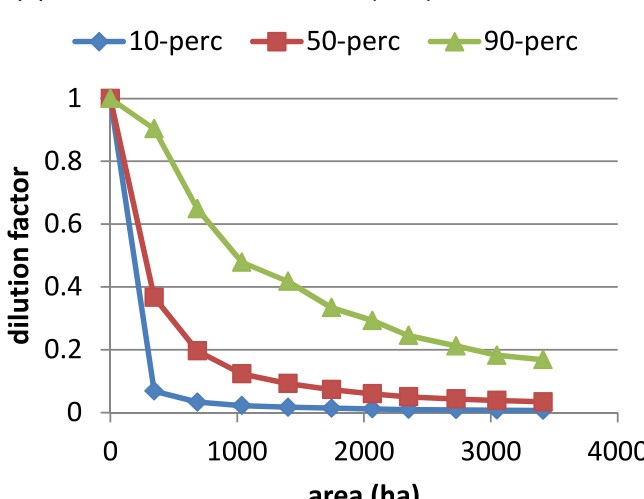

(B) "Off-field Habitats" medium quality

**Figure 5 On an area-base flower strips are more effective than off-field patches.** To obtain the same degree of dilution, much more area is required of managed off-field habitat patches (B), compared to flower strips (A) with identical properties. This is related to landscape characteristics and a consequence of off-field patches being in general further away from target crop field and hive location.

on our model. These alternative patches need to provide resources that are at least equally (RL-model) or more (SO-model) attractive.

The structure of the landscape determined to a large extent the number of daily exploited resource patches. In a coarse-grained landscape with few large resource patches of mass-flowering crops the use of a single to a few patches was predicted by the SO- and the RL-model, respectively. In landscapes with many small resource patches, however, results differed considerably between the two model versions. The SO-model could in theory predict up to 10 different patches to be exploited, one for each hour. However, this was not observed, as switching each hour to a new resource would require strong resource depletion to take place. The RL-model could for the same landscape predict up to 75

patches accounting for in total 90% of the daily nectar collection. Assuming competition with other nectar-foraging species or other colonies (Fig. S11 compared to Fig. S8) indeed increased the number of exploited patches for the SO-model because of earlier depletion of the selected patch, while it decreased for the RL-model presumably because the total set of attractive patches over the day became smaller.

For a colony of feral honeybees in temperate deciduous forest, on average 9.7 ($\pm$4.9) resource patches (pollen and nectar) accounted for 90% of the daily foraging activity (*Visscher & Seeley, 1982*). For two colonies of African honeybees in the Okavango River Delta, *Schneider (1989)* found that 16.2 ($\pm$9.6) and 17.5 ($\pm$5.9) sites/day accounted for 90% of the daily foraging activity on pollen and nectar. In both studies a continuous day-to-day redistribution of foragers over patches was observed, with many more patches being present than used each day. Beekman and colleagues *(2004)* found for two large and two small colonies of honeybees in an urban environment on average 15.8 and 15.0 patches being used per day (nectar and pollen), respectively.

These observations indicate that less optimization might happen than assumed in the SO-model, but also that less dilution of foraging effort is occurring than predicted by the RL-model for landscapes with many resource patches.

## Landscape level exposure dilution

In current agro-ecosystems few high quality off-field resource patches will be available to honeybees (*De la Rúa et al., 2009*). The energetic model predicts that mass-flowering crops will be selected even when these are located at a large distance from the hive. Observations indeed indicate that in the period during which mass-flowering crops are available, these crops constitute the predominant nectar source (*Requier et al., 2015*). The off-field resource patches that are present will under these conditions experience pollinator dilution (*Holzschuh et al., 2011*), i.e., reduced wild plant pollination because OSR fields are preferred. Depletion is not likely to play a large role in these crop-dominated systems, as most crop fields are large and flower densities are in the saturating range of the functional response. The model thus suggests that in general there will be little potential for dilution of exposure in such landscapes, because the only resource that can be as attractive as a mass-flowering crop is another potentially sprayed mass-flowering crop.

We tested three mechanisms for exposure reduction at landscape scales. Starting from the worst-case assumption that the nearest target crop field was always treated, all tests indicated that there were only chances for dilution when uncontaminated patches were equally or more attractive than the target crop, nearby, numerous and of sufficient size. In a landscape with only crop fields providing resources, dilution appeared unlikely because the probability of having a more attractive uncontaminated field in the neighbourhood was simply too small, even though this neighbourhood became extended when sugar content was higher. Flower strips had the highest potential for dilution at the beehive: when all target crop fields were surrounded by some flowers strips the 90-percentile dilution factor became small for almost all sites. Off-field habitat patches were more than one order of magnitude less efficient compared to flower strips: around 30 times as much managed area was needed to achieve the same dilution. The advantage of flower strips was that they

were located within the target crop field, so they easily met the small distance condition. On the other hand, such a location in a target crop field may pose an additional exposure risk when crop treatment also causes exposure in the strips. With spray drift, effectiveness in attracting foragers may trade off against higher exposure risk for flower strips, and it might be possible to achieve a higher dilution for off-field patches that are carefully chosen with respect to their location relative to crop fields: close enough to attract foragers, but far enough to avoid contamination.

## Model complexity and limitations of the predictions

The current application of the developed model was not meant to deliver accurate predictions of absolute real-world nectar concentrations at the beehive. The purpose was instead to calculate the dilution factor as a relative measure of the effect of alternative nectar sources on worst-case nectar concentration estimates. The model equations as presented in this manuscript are in principle realistic enough to deliver absolute exposure estimates at landscape levels. This could, however, only be achieved with using more realistic landscape information of higher quality.

For the foraging model, realism means that the presence and status of all relevant sources of nectar in the landscape need to be known. When these data were available, predicted foraging locations could be compared to empirical data as for example obtained from dance analyses (*Couvillon, Schurch & Ratnieks, 2014a*). Regarding exposure, realism means that all applications of a pesticide in the considered region have to be taken into account, and the contamination not only of the treated crops but also of in-field flower strips and off-field patches of natural habitats providing nectar flowers has to be considered. Such real-world spatially-explicit exposure would be complex, and dependent on application patterns and the fate of pesticides in the field. For instance, exposure to systemic insecticides was reported to be even larger in wildflower patches than in the actual crop on which they were applied (*Botias et al., 2015*).

Colony dynamics as e.g., explicitly simulated in the BEEHAVE model (*Becher et al., 2014*) were not considered. Focussing on exposure resulting from foraging allowed us instead to avoid the complexity and the high data requirements of full colony models and made it feasible to apply the model in assessments at large spatial scales. The model dealt with nectar foraging only, as pollen foraging follows different rules. Pollen sources may not differ as much in quality as nectar sources, and net energetic efficiency does not play a role. A colony's annual need of pollen (protein) is smaller than that of nectar (25 compared to 125 kg (*Seeley, 1985*)) and the overall sugar consumption of individual bees exceeds the protein consumption about 5 times (*Rortais et al., 2005*). However, there are good reasons also to address exposure via pollen and to develop a landscape-level pollen foraging model to be used in combination with our nectar foraging model. For instance, systemic insecticide concentrations may be higher in pollen, pollen is consumed by nurse bees that may constitute a critical, sensitive subset of the colony population, and pollen is directly stored, unmixed, and thus potentially preserves any high doses that might occur.

We applied the energetics-based model in a theoretical analysis of mitigation options, based to a large extent on real-landscape data. The results show the potential of using the

model in a large-scale risk assessment and give general insight in the type of mitigation strategy that has the highest chance to be effective.

### Funding

The research presented is conducted in the context of research program BO20-002-011 of the Dutch Ministry of Economic Affairs. The funders had no role in study design, data collection and analysis, decision to publish, or preparation of the manuscript.

### Grant Disclosures

The following grant information was disclosed by the authors:
Dutch Ministry of Economic Affairs: BO20-002-011.

### Competing Interests

The authors declare there are no competing interests.

### Author Contributions

- Johannes M. Baveco conceived and designed the experiments, performed the experiments, analyzed the data, contributed reagents/materials/analysis tools, wrote the paper, prepared figures and/or tables.
- Andreas Focks conceived and designed the experiments, contributed reagents/materials/analysis tools, wrote the paper, reviewed drafts of the paper.
- Dick Belgers, Jozef J.M. van der Steen, Jos J.T.I. Boesten and Ivo Roessink contributed reagents/materials/analysis tools, reviewed drafts of the paper.

### Data Availability

The raw data has been supplied as Supplemental Information.

### Supplemental Information

Supplemental information for this article can be found online at http://dx.doi.org/10.7717/peerj.2293#supplemental-information.

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
