# Peer review of "An energetics-based honeybee nectar-foraging model used to assess the potential for landscape-level pesticide exposure dilution"

_PeerJ, doi:10.7717/peerj.2293_

## Round 0.1 · original submission · Major Revisions

Reviewer 2 expresses serious concerns about some of the assumptions in the model. The authors need to pay special attention to these concerns and I strongly recommend to conduct sensitivity analyses on key variables. The authors also need to specify source code, software and GIS data as recommended by reviewer 1.

·

Basic reporting

General comments:

The authors must specify the software with which they implemented their model and provide the source code for review. Authors should also include the GIS files they used to define their landscape.

The manuscript is generally well-written, but there are minor grammatical errors throughout, mostly regarding the use of commas. I also think the impact of the manuscript would be enhanced considerably if its length could be reduced through more concise style. I would recommend having someone with expertise in English grammar and style contribute to the revision process.

In several places, I advise the authors to engage more thoroughly or more carefully with the literature (detailed below).


Line-by-line comments:

8 (of abstract body): I would say “patch”, not “field”, since this applies to both mass flowering crop fields and off-field resource patches, right?

2-3: Is there evidence that honey bees are declining in agricultural landscapes? There is plenty of evidence that agricultural intensification threatens non-Apis bees, but honey bees have actually been shown to be quite robust to agricultural land use, as you note later in this paragraph. A quote from Potts et al. (2010): “In both quantitative reviews [Ricketts et al. (2008) and Winfree et al. (2009)], no such effects [of distance from natural habitat] were found on honey bees, which occurred as managed species in many of the studies considered.”

5-8: For more on the robustness of honey bees to agricultural land use, see the following papers and references within:
Winfree et al. (2009). A meta-analysis of bees’ responses to anthropogenic disturbance. Ecology.
Ricketts et al. (2008). Landscape effects on crop pollination services: are there general patterns? Ecology Letters.

10-14: Can this paragraph be omitted? It does not seem to add anything to the flow of thought, since the authors' project is not a mechanistic effects model in the sense that BEEHAVE is. BEEHAVE is concerned with modeling in-hive effects on things like population structure, disease rate, food storage, etc. It could be modified to take pesticide exposure as an input, but it does not model exposure per se, which is what the authors' model does. If the authors want to talk about BEEHAVE, I recommend they do so in the Discussion section, where they can make the good point that the output of their model could become the input to BEEHAVE to create a powerful combination of exposure modeling and mechanistic effects modeling.

15-19: Somewhere in the introduction, and probably right about here, the authors should discuss this paper:
Carreck and Ratnieks (2014). The dose makes the poison: have “field realistic” rates of exposure of bees to neonicotinoid insecticides been overestimated in laboratory studies? J. of Apic. Res.

21-27: This discussion should not be based on Garbuzov et al., which was not primarily designed to study foraging distance. Couvillon et al. (2014, 2015) are the most thorough studies of honey bee foraging distance, and they should provide the foundation for any discussion of spatial scale vis-a-vis honey bee exposure:
Couvillon et al. (2014). Waggle Dance Distances as Integrative Indicators of Seasonal Foraging Challenges. PLoS ONE.
Couvillon et al. (2015). Honey bee foraging distance depends on month and forage type. Apidologie.

107: What does “NB” mean in this sentence? Just a typo?

465-477: It’s fine that the authors focus only on nectar in their model, but I would caution them not to be dismissive of the importance of exposure via pollen. First of all, the systemic presence of neonicotinoids tends to be higher in pollen than in nectar. Secondly, the nurse bee population that is directly exposed to pollen-borne pesticide is the most critical subset of the colony population due to their young age, high resource investiture, brood care responsibilities, and unique pollen digestion ability. Third, nectar gets mixed extensively prior to consumption, averaging out contamination levels from different sources. Pollen is consumed unmixed, preserving high doses from dilution. Just something to consider.

664-665: Please cite the full peer-reviewed version of Sponsler and Johnson (2015) rather than the preprint:
Sponsler D, Johnson R. (2015) Honey bee success predicted by landscape composition in Ohio, USA. PeerJ 3:e838 https://doi.org/10.7717/peerj.838

Experimental design

General comments:

My biggest concerns are about the values of floral density and nectar concentration assumed for floral strips and off-field clover. I would ask either that the chosen values be well-supported from the literature, or that a range of plausible values be subject to sensitivity analysis.


Line-by-line comments:

139-140: How well does the authors' model actually reproduce this observation under each of the conditions under which they run it? That is, how many patches are visited per day under each set of conditions? I would be interested in seeing some reporting on this in the results or discussion.

143-145: I don’t understand this sentence. Could this be rephrased or elaborated?

179-191: When the authors run the model in their real landscape to estimate exposure under different scenarios, do they assume consumption by foragers or no consumption? And do they assume metabolism of consumed pesticide or no metabolism? I would like to see this made more explicit.

252-253: It's not clear to me how the authors implement their model to achieve the exhaustive iteration they discuss here. Source code should be provided.

298-299: Is this assumption of equal floral density and 50% higher sugar content in flower strips compared to OSR justifiable from the literature?

315-318: The realism of these floral density settings seems suspect. See my comment below on Table 2.

Table 2: Can the use of 6000 and 9000 clover flowers per square meter be justified? That works out to 150 and 225 flower heads per square meter, respectively, assuming the maximum of 40 flowers per head. Is that realistic?

Validity of the findings

517-518: This is because the nearest field was always assumed to be treated, right? One might find a very different result if treatment were assigned probabilistically to all fields, including the nearest. I understand the authors' choice of conservative assumptions, though.

Additional comments

Your project seems very well-conceived and well-executed, and I consider it a very timely contribution to the field of honey bee toxicology and risk assessment. The only major barrier to publication, in my opinion, is that you have not provided your source code for review.

Reviewer 2 ·

Basic reporting

The manuscript is very well written and was easy to read, which is impressive considering the level of complexity of building the model. I did not like how the first mention of the crops used was not reported until the "model analysis and applications" section. Figure 7 has no Y-axis label, so it is not immediately obvious what the figure is depicting.

Experimental design

Please see my comments to the author, but I feel that generally there are too many assumptions in the model, and it is too simple to be of use in its current form.

Validity of the findings

The authors have acknowledge themselves in the conclusion that detailed GIS data and resource specific values need to be collected before the model can be used for any form of risk assessment. This is why I think that at the moment, the model is not robust enough; these values are critical.

Additional comments

This is a very well written manuscript and the development of the model is well described and methodical. However, I do not feel comfortable recommending its publication in its current stage of development. The variables included in the model are all necessary to understand how foraging patterns will shift, however I feel that many of the underlying assumptions are far too over simplified. For example, I don’t think the authors can assume symmetry breaking will always occur between multiple resources. Honey bees always foraging on a single resource at any time is far too simplistic; many resources will be exploited simultaneously and which then dramatically changes the parameters of the model. The complexities of the nectar energetics is also too simple. I support the inclusion of Oil Seed Rape and Clover as the target crops here, considering how commonly they are grown across Europe, but all the other landscape variables that go alongside growing such crops need to be considered before a good idea of how bees will forage between the two can be predicted. A larger variety of crops and wild flowers, nectar concentrations, floral density measures, flower attractiveness, speed of nectar replenishment and the plethora of variables that go alongside these should be considered and the model developed accordingly. What about other pollinators affecting nectar replenishment? Honey bees also forage for water, and their water foraging will be dependent on various interacting factors, as will the nectar concentration, this will shift the forager allocation to different resources. Why are the floral strips assumed to be 50% higher sugar content than OSR, what is this based on? Are 9000 flowers per meter realistic for clover? It seems to be a very high number! I also could not see where the authors decided whether the pesticide was metabolized or not. They suggested that it may or may not be, so what if it is? The effect of the pesticide on the bee’s behaviour then needs to be incorporated into the model. We know that individual behaviour of bees is impacted by pesticides, so this also needs to be taken account of when bees are foraging on sprayed fields and how they might redirect themselves following exposure. Which pesticides? At what concentration? A combination of how many pesticides? These are just a few examples of why I think that the model is not yet ready for publication. I believe that the model will be useful for predicting the patterns of exposure dilution, and I agree that understanding dilution effects is important problem to explore. It’s an extremely complex problem though, and I do not think the model is anywhere near developed enough to be ready to address it.

---

## Round 0.2 · accepted · Accept

The authors have done a very thorough job revising their manuscript. Well done!